# Balance dynamics are related to age and levels of expertise. Application in young and adult tennis players

**Carla Caballero**[ID]**, David Barbado, Héctor Hérnandez-Davó, José Luis Hernández-Davó, Francisco J. Moreno***

Sports Research Centre, Department of Sport Sciences, Miguel Hernández University of Elche, Elche, Alicante, Spain

* fmoreno@umh.es

**Data Availability Statement:** All relevant data are within the manuscript and its Supporting information files.

## Abstract

In tennis, coaches consider balance fundamental for the acquisition of skilled motor performance. However, the potential relationship between balance and tennis expertise and performance has not been explored yet. Therefore, this study assessed the relationship between balance and tennis performance using linear and non-linear parameters through 1) the comparison of tennis players of different ages and levels of expertise, and 2) analyzing the relationship between balance and tennis serving speed and accuracy. One hundred and six recreational and expert male tennis players took part in the study (age range 10–35 years old). Temporal dynamics of postural control during a balance task on an unstable surface were analyzed through the mean velocity and the detrended fluctuation analysis ($DFA_V$) of center of pressure (COP). Tennis serve performance was quantified by measuring accuracy and speed. Traditional variables measuring balance performance only showed differences according to age but not to sport performance. COP showed a reduction of auto-correlated variability (reflected by $DFA_V$) with age but mainly in expert players. COP dynamics was the only balance parameter discriminating sport expertise and it was related to age. Balance dynamics exhibited by expert tennis players $DFA_V$ results support the idea that, along the years, sport experience induces balance adaptations characterized by a higher ability to perform postural adjustments. These results also reinforce the use of non-linear analysis to reveal subtle balance adaptations produced by sport practice. Finally, the lack of correlations suggests that balance, measured with scattering variables, in a non-specific task is not a main determinant of sport performance in tennis serve.

## Introduction

Balance is one of the main features of motor control as every human movement involves it. Balance has been defined as the ability of the neuromuscular system to integrate visual, vestibular and somatosensory feedback to control the body's center of gravity projection [1, 2]. Therefore, balance control has been frequently analyzed by the use of tasks that challenge the

**Funding:** This study was made possible by financial support from the Economy, Industry and Competitiveness Ministry of Spain, projects cod DEP2013-44160-P and DEP2016- 79395-P, Spanish Government. Author who received the award: Francisco J. Moreno The funders had no role in study design, data collection and analysis, decision to publish, or preparation of the manuscript.

**Competing interests:** The authors have declared that no competing interests exist.

individual in maintaining the position of the body's center of gravity vertically over the base of support [1, 2], and it has been commonly assessed through the body's postural sway [3].

Specifically, in sport training programs balance control is frequently targeted due to its role as a preventive factor for reducing injury rates as well as for increasing sports performance [1]. In this sense, the relationship between balance and sport has been shown to be bidirectional, and performing sports activities is considered a meaningful way to improve balance control [4]. Sport experience seems to induce structural and functional adaptations that could reduce the balance deterioration caused by age, pathologies, etc. [5, 6]. Nevertheless, the magnitude and the direction of the balance adaptations seems to depend highly on the features of the athletic skills involved in the different sports [1]. Commonly, athletes from sports in which keeping balance is directly related to performance, like gymnastics, have shown a higher balance compared to their less proficient counterparts or sedentary (non-athlete) individuals [7, 8]. However, these adaptations have not always been observed in other sports [9, 10].

In tennis, coaches consider balance as a fundamental capability for the acquisition of skilled motor performance, being usually aimed at specific training or rehabilitation programs [11]. Tennis skills compromise complex stroke movements as well as short-burst and multidirectional intermittent actions, including sprinting, fast cutting maneuvers, sudden decelerations, and pivoting maneuvers, which put the tennis player under physical stress [12]. According to these physical and technique demands, having appropriate balance levels would be a primary determinant for tennis success as they would not only prompt better drill performance [13, 14] but they would also reduce the risk of suffering lower extremity injuries, such as muscle strains and ligament sprains [15]. In addition, it would also be expected that tennis experience induces balance adaptations. Nevertheless, there is little evidence confirming both assumptions. To the best of authors' knowledge, only one cross-sectional study comparing a small sample of tennis players (n = 12) against controls, has suggested that tennis experience induces balance adaptations [16]. However, no study has assessed the potential relationship between balance and tennis performance, either by comparing different cohorts of tennis players with different levels of expertise (i.e., expert vs. novice), or by analyzing the relationship between balance and specific tennis skills.

In order to verify the potential link between balance and expertise level, and in particular in tennis performance, two potential issues should be addressed. On the one hand, the balance task must be challenging enough, according to the features of the target population, in order to reveal the real balance status of each individual [17]. Previous studies in surfers have suggested the need of challenging conditions to better address the balance differences between skilled and non-skilled athletes [18, 19]. Based on this idea, unperturbed bipedal stance would not to be sensitive enough to detect subtle differences induced by sport experience [4]. That way, more demanding tasks (e.g. unstable surfaces, reducing the base of support, performing a dual-task, etc.) would be needed to show the real balance performance of tennis players.

On the other hand, traditional scattering parameters – such as the root mean square, the resultant distance, the center of pressure sway area, or the mean velocity—have been used to describe the sway and the dispersion or the area during a given time with a balance task [20]. However, a recent approach derived from the theory of stochastic dynamics, suggests that non-linear parameters should also be implemented to reveal the true state of the postural balance control system [21, 22]. Specifically, non-linear parameters (e.g., detrended fluctuation analysis), which analyze how motor behavior changes over time, have been linked to some potential underlying mechanisms related to a higher or lower ability to perform motion adjustments [23, 24]. For example, DFA scores have been related to the individuals' sensitivity to their own motor error [21] or the effort to accomplish a motor task [23]. Analyzing tennis

players' balance behavior through non-linear parameters would help to reveal if tennis experience is related to higher flexibility to perform postural adjustment during balance tasks.

Based on the above-mentioned rationale and taking into account that: i) balance control has been related to motor control performance regarding age [24–27] or sport skills [1, 28]; and that ii) balance has been considered by coaches as one of the most important determinants of serve success [29], this study explored if tennis players exhibited better balance control according to their sport experience. Therefore, the main aim of this study was to assess the relationship between balance and tennis performance through 1) the comparison of tennis players of different ages and levels of expertise, and 2) analyzing the relationship between balance and tennis serving speed and accuracy. Tennis players' balance was assessed using an unstable balance task and analyzing traditional and non-linear parameters in order to explore the dynamics of the postural adjustments and to maximize the potential differences induced by sport experience. Tennis performance was assessed through the accuracy of the tennis serve as a way to analyze a tennis skill very relevant to tennis success but not directly related to balance control.

## Materials & methods

### Participants

One hundred and six male tennis players took part in this study. Players had different levels of experience and frequency of practice (see Table 1 for demographic information). Fifty-five of them were expert players recruited from the Royal Spanish Tennis Federation during the 2012–2013 sport season. In order to be recruited, they had to meet the following criteria: to train at least four times per week; and to play in national or international competitions. In addition, fifty-one participants were recruited as recreational players in the same season from the university where the study was conducted. The recreational players met the following criteria: i) to train no more than two times per week; and to have not taken part in any national or international competition. Apart from their level of expertise, participants were also categorized according to their age [under 12 years old (U12); under 16 years old (U16), older than 18 years old)]. None of them had any previous experience in the balance task used in this study.

Written informed consent was obtained from each participant before testing. For both U12 (under 12 years old) and U16 (under 16 years old) groups, written informed consent was signed by the players' guardians. Data were treated anonymously, and all participants were informed of the risks and benefits of the trial. The experimental procedures used in this study

**Table 1. Demographic information from the participants who took part in the study.**

| | TENNIS PLAYERS | | | | | |
| | *Expert | | | #Recreational | | |
| | N | Age | Height | N | Age | Height |
|---|---|---|---|---|---|---|
| **U12** | 22 | 11.84 (1.12) | 1.60 (.11) | 14 | 11.58 (.81) | 1.59 (.07) |
| **U16** | 21 | 14.61 (.91) | 1.76 (.04) | 19 | 15.24 (1.32) | 1.69 (.05) |
| **+ 18** | 12 | 23.00 (6.18) | 1.81 (.04) | 18 | 22.67 (1.76) | 1.76 (.06) |

Data are provided as mean (standard deviation).

*The participants who were considered expert players were those who met the following criteria: i) they had been selected by the Spanish Tennis Federation; ii) they trained at least 4 times per week; iii) they played in national or international competitions.

# The participants who were considered recreational players were those who met the following criteria: i) they trained no more than 2 times per week; ii) they had not taken part in any national or international competition.

U12 = under 12 years old; U16 = under 16 years old; + 18 = older than 18 years old.

were in accordance with the Declaration of Helsinki and were approved by the Office for Research Ethics of the Miguel Hernandez University of Elche (Ref: DPS.FMH.01.13 and DPS. FMH.01.16).

## Experimental procedure and data collection

Participants performed two different protocols: 1) balance task; and 2) tennis serve.

**1) Balance task protocol.** Participants were asked to stand 'as still as possible' on an unstable surface. The unstable surface consisted of a rigid wooden platform (diameter: 55 cm) affixed to the flat surface of a polyester resin hemisphere (diameter of the hemisphere: 35 cm; height of the platform relative to the bottom of the hemisphere: 12 cm). Their feet were placed shoulder-width apart and their hands rested on their hips. The line between the participants' heels had to match with the medial-lateral axis of a force platform (Kistler, Switzerland, Model 9286AA). Participants performed one trial keeping this position for 60 seconds [18, 30, 31]. In order to assess postural stability, the unstable surface was placed on a force platform. Ground reaction forces were recorded (1000 Hz) to analyze the center of pressure (COP) data that were subsampled at 20 Hz to avoid artificial co-linearities [32].

**2) Tennis serve procedure.** For all the evaluations, the players placed themselves behind the baseline on the left-hand side of a clay tennis court, at a distance of 0.8 m from the center service mark. Each player executed 10 min of specific warm-up before the evaluation, including three serves. The protocol consisted of 30 serves, distributed in three blocks of 10 serves, aiming at a square target located in the service box (Fig 1). There was a 30 s rest between blocks and five seconds between serves. We asked the participants to 'serve at the highest speed and with the highest accuracy possible, aiming at the target' [33]. The balls were handed over one by one for each serve (Brilliance tennis balls, Dunlop, Shirebrook, UK).

In order to assess the tennis serve performance, a sports radar (SR3600, Homosassa, FL, USA) was used to collect the maximum speed of the ball in each serve (sensitivity: ± 0.44 m/s). The radar was placed behind the players pointing at the target located in the service box. To quantify the serve accuracy, the bounce of the ball was digitalized using a Sony HDR-SR8E

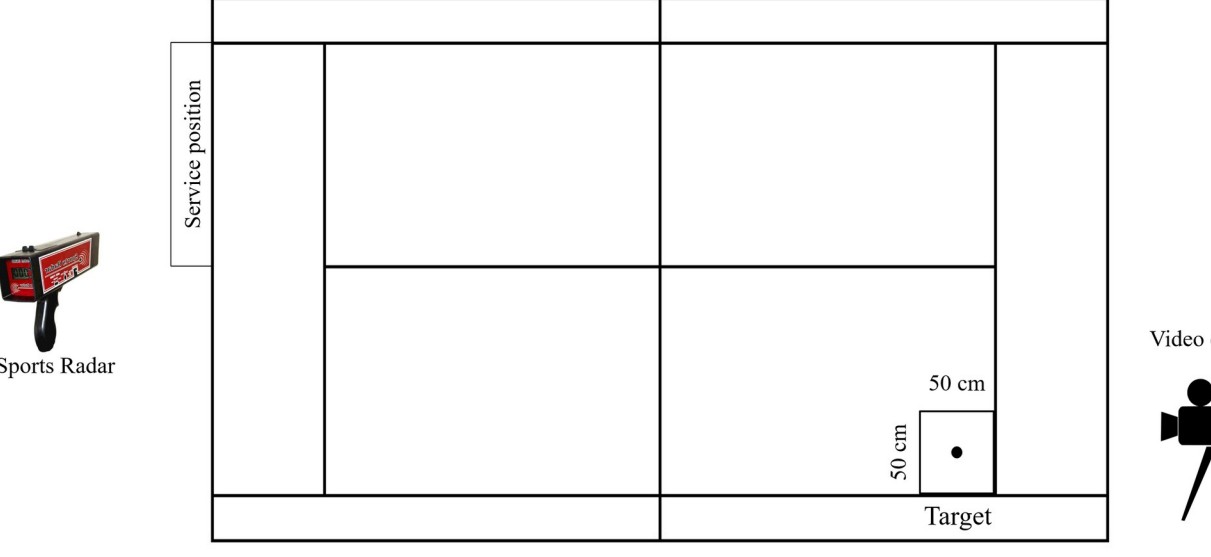

**Fig 1. Tennis court and instruments distribution.**

digital camera (Sony, Tokyo, Japan) (50 Hz sampling frequency). The camera was placed behind the service box, at a 3 m height from the floor, pointing to the service box (Fig 1).

## Data analysis and reduction

**1) Balance analysis.**  Prior to the analysis, the first 10 s of each trial were discarded to avoid non-stationarity related to the start of the measurement [34]. The length of time series analyzed was 1000 data. A low-pass filter (4th-order, zero-phase-lag, Butterworth, 5 Hz cut-off frequency) was performed, according to Lin, Seol [31].

The mean of the velocity magnitude ($MVM_{COP}$; mm/s), of the center of pressure was computed according to Prieto, Myklebust [20], and used as a representative linear variable to characterize balance performance, as it has been shown to be the most reliable traditional parameter [32, 35]. To assess the non-linear dynamic of the COP, we calculated the detrended fluctuation analysis ($DFA_V$) from the velocity magnitude time series. For more information about this procedure, please, review Caballero, Barbado [32]. $DFA_V$ represents a modification of the classic root mean square analysis with a random walk to evaluate the presence of long-term correlations within a time series using a parameter referred to as the scaling index α [36, 37]. The scaling index α corresponds to a statistical dependence between fluctuations at one timescale and those fluctuations over multiple timescales. This procedure assesses the extent to which further motor behavior is dependent on previous fluctuations, describing the complexity of the time series [38]. Less dependency on previous behavior (lower long-range auto-correlation; lower α) has been interpreted as a higher flexibility to perform motion adjustments [39]. This measure was computed according to the procedures of Peng, Havlin [37]. In this study, the slope α was obtained from the window range $4 \leq n \leq N/10$ to maximize the long-range correlations and reduce errors incurred by estimating α [40], being N the total number of the time series data point. Different values of α indicate the following: $α > 0.5$ implies persistence in position (the trajectory tends to remain in its current direction); $α < 0.5$ implies anti-persistence in position (the trajectory tends to return to where it came from) [41].

**2) Tennis serve analysis.**  The speed of the ball (km/h) was computed from the maximum speed across trials. In order to compute the accuracy of the serve, the mean radial error (MRE) of the ball's bounce (m) was calculated. It was measured as the average of absolute distance to the center of the target (we considered the center of the square target as the aiming point) (see Fig 1). Ball bounces were video recorded and digitalized. Real-space Cartesian coordinates were computed by a MATLAB routine (V.7.11. MathWorks, Natick, MA).

## Statistical analysis

Normality of the variables was evaluated using the Kolmogorov-Smirnov test with the Lilliefors correction. Two-way independent measures ANOVAs were carried out to assess between group differences in all parameters, being age (U12, U16 and +18) and skill level (*expert* and *recreational*) as the between-subjects' factors. Partial eta-squared ($\eta_p^2$) was calculated as a measure of effect size and to provide a proportion of the overall variance that is attributable to the factor. Values of effect size $\geq 0.64$ were considered strong, around 0.25 were considered moderate and $\leq 0.04$ were considered small [42]. Bonferroni adjustment for multiple comparisons was performed to ascertain differences between each specific subgroup. Finally, Pearson product-moment correlation coefficients were calculated to assess relationships between balance performance variables ($MVM_{COP}$ and $DFA_V$) and sport performance variables (accuracy and ball speed) in each specific subgroup. Statistical tests were made using IBM SPSS Statistics 22.0.0.0. The alpha value of significance effect was set at $p < 0.05$.

**Table 2. Correlations between all the variables for all the subjects.**

|  | Ball speed | MVM$_{COP}$ | DFA$_V$ |
|---|---|---|---|
| **MRE** | **-0.611**\*\* | 0.068 | **0.225**\* |
| **Ball speed** | | -0.146 | **-0.255**\*\* |
| **MVM$_{COP}$** | | | -0.089 |

\* p < 0.05;

\*\* p < 0.01;

**bold font** is used for all the significant correlations.

## Results

Average values obtained for each age range and expertise level groups in all variables are shown in Table 2. The whole database can be found in the Supporting Information (S1 Database). Main effect of expertise was found on accuracy ($F_{1,105}$ = 73.939; p < 0.001; $\eta_p^2 = 0.413$) and ball speed ($F_{1,105}$ = 116.264; p < 0.001; $\eta_p^2 = 0.525$). Expert players served with better accuracy and showed a higher ball speed than recreational players despite the age of the player. In addition, a significant improvement in serve accuracy ($F_{2,105}$ = 8.042; p = 0.001; $\eta_p^2 = 0.133$) and ball speed ($F_{2,105}$ = 44.989; p < 0.001; $\eta_p^2 = 0.461$) was observed as the age of the players increased.

Concerning the balance task, main effect of age was found, showing a reduction in both MVM$_{COP}$ ($F_{2,105}$ = 14.378; p < 0.001; $\eta_p^2 = 0.215$) and DFA$_V$ ($F_{2,105}$ = 3.420; p = 0.036; $\eta_p^2 = 0.061$) in the +18 group. This group showed significant lower COP velocity in both, expert and recreational players and only the expert groups displayed lower DFA$_V$ too (Fig 2). Balance differences between expert and recreational players were only observed in DFA$_V$

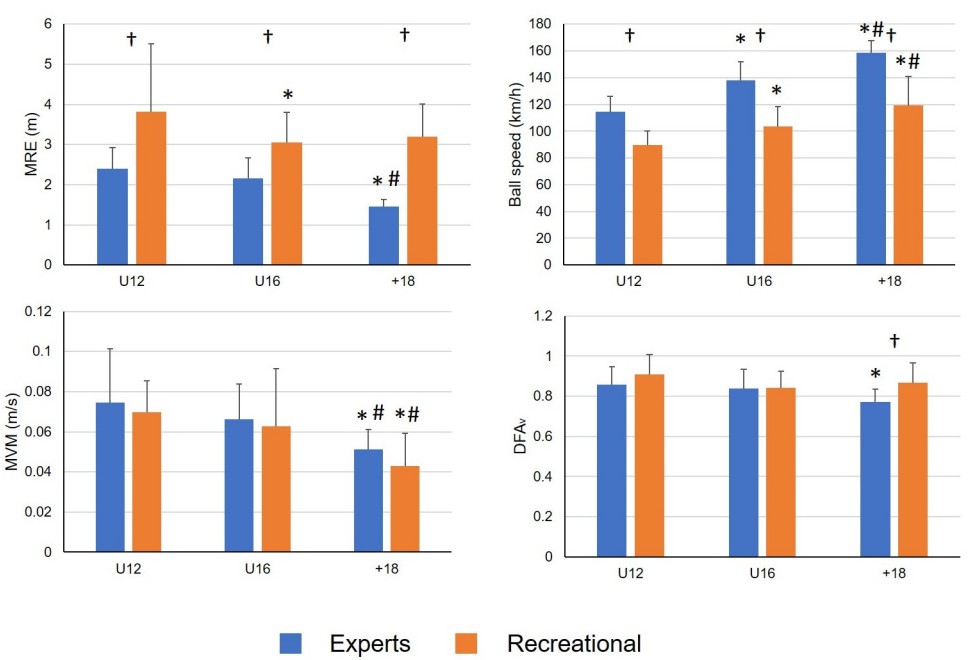

**Fig 2. Pair comparisons between age ranges and expertise levels.**

**Table 3. Correlations between all the variables for expert players.**

|  |  | Ball speed | MVM$_{COP}$ | DFA$_V$ |
|---|---|---|---|---|
| U12 | MRE | -0.316 | -0.022 | -.0372 |
|  | Ball speed |  | -0.040 | 0.083 |
|  | MVM$_{COP}$ |  |  | -0.164 |
|  |  | Ball speed | MVM$_{COP}$ | DFA$_V$ |
| U16 | MRE | -.0575** | **0.461*** | -0.151 |
|  | Ball speed |  | -0.090 | 0.190 |
|  | MVM$_{COP}$ |  |  | -0.130 |
|  |  | Ball speed | MVM$_{COP}$ | DFA$_V$ |
| +18 | MRE | -0.379 | 0.216 | -0.062 |
|  | Ball speed |  | -0.052 | 0.247 |
|  | MVM$_{COP}$ |  |  | **-0.599*** |

* $p < 0.05$;

** $p < 0.01$;

**bold font** is used for all the significant correlations.

($F_{1,105} = 6.596$; $p = 0.012$; $\eta_p^2 = 0.059$). Multiple pair comparisons showed that the differences in DFA$_V$ between expert and recreational players were significant in the +18 group (Fig 2).

Once the differences between age ranges and skill levels were analyzed, Pearson product-moment correlation coefficients were computed to assess the relationship between the serve performance (MRE and ball speed variables) and balance parameters. The correlational analysis of the whole sample (Table 2) reflects the main results reflected in ANOVA, showing an inverse relationship between ball speed and MRE ($r = -0.611$; $p < 0.001$). There was no relation between balance performance and serve performance but DFA$_V$ values were related to both MRE ($r = 0.225$; $p = 0.02$) and ball speed ($r = -0.255$; $p = 0.008$). Overall, no systematic correlation was observed between balance and serve parameters in expert (Table 3) nor in recreational players (Table 4). Only the expert U16 group showed a positive relationship between MVM$_{COP}$ and serve accuracy ($r = 0.461$; $p = 0.036$).

**Table 4. Correlations between all the variables for recreational players.**

|  |  | Ball speed | MVM$_{COP}$ | DFA$_V$ |
|---|---|---|---|---|
| U12 | MRE | -0.615* | -0.001 | 0.158 |
|  | Ball speed |  | -0.116 | -0.272 |
|  | MVM$_{COP}$ |  |  | -0.061 |
|  |  | Ball speed | MVM$_{COP}$ | DFA$_V$ |
| U16 | MRE | -0.215 | 0.377 | 0.357 |
|  | Ball speed |  | 0.112 | -0.247 |
|  | MVM$_{COP}$ |  |  | -0.121 |
|  |  | Ball speed | MVM$_{COP}$ | DFA$_V$ |
| +18 | MRE | -0.182 | -0.148 | -0.059 |
|  | Ball speed |  | -0.073 | 0.054 |
|  | MVM$_{COP}$ |  |  | -0.329 |

* $p < 0.05$;

**bold font** is used for all the significant correlations.

## Discussion

In the present study, we analyzed the relationship between balance and tennis performance through the comparison of different cohorts of tennis players with different levels of expertise, and through the analysis of the relationship between balance and tennis serve performance.

As it was expected, the first results of this study showed significant differences in performance variables, error (MRE) and ball speed between expert and recreational players. For all the age ranges, expert players showed higher accuracy (lower MRE) and ball speed. These results confirmed that the classification of the sample according to skill levels was appropriate, thus the rest of the analysis are valid to assess the relationship between balance and tennis performance. Regarding the age ranges, older players showed higher ball speed and higher accuracy in both groups, expert and recreational.

Regarding balance performance, there was a significant reduction of the body sway (i.e., lower COP velocity) according to the age but not according to the expertise level. The lower COP velocity values ($MVM_{COP}$) observed in our study due to the age are in line with previous findings, which suggested that body sway reduction is mainly related to the development of the different afferent sensory systems caused by age [43, 44]. Conversely, in our study, we did not observe a decrease in body sway caused by sport expertise. These results are in line with previous works which did not find any significant effect of sport experience in balance adaptation when traditional postural sway parameters were used, even in sports that are so balance demanding like surfing or judo [10, 18]. This could be interpreted as that those traditional parameters are not sensitive enough to detect balance adaptations caused by sport specialization and, therefore, the use of other indexes is required. This hypothesis seems to be supported by the fact that, in our study, higher balance complexity (i.e., lower $DFA_V$) was indeed observed not only according to age but also according to the level of specialization. This is, players with better tennis performance (the oldest players in the expert players group) showed less auto-correlated COP velocity time series. Therefore, these results reinforce that $DFA_V$ was more sensitive than balance body sway parameters to quantify how motor behavior changes over time, providing information about changes in postural control and the system ability to adapt that linear variables cannot identify [21, 22]. According to several works, the low $DFA_V$ in the COP excursion revealed that expert and oldest tennis players performed a higher number of movement adjustments during the balance task [25]. These results could indicate that these participants displayed a more exploratory behavior [45] or higher sensitivity to their own motor error [21], allowing them to obtain better performance and facilitating adaptation processes [21, 30]. The fact that only expert players showed a reduction in $DFA_V$ scores with age while no changes were observed in the recreational tennis players would reinforce that tennis experience rather than age seems to be related to a better postural control system, characterized by a higher ability to perform motor adjustments.

Regarding the correlational analyses, when the whole sample was assessed, a direct relationship between accuracy and velocity was observed. Previous experiments in tennis serve have found this apparent exception to Fitts' law [46], so the better tennis players are able to serve faster and more accurately (lower MRE). However, these results are mainly due to the effect of *age* and *expertise*, as the ANOVAs showed. The fact that older recreational players did not exhibit significantly greater accuracy than younger players would support that expertise is based not only on age but also on deliberated practice combined with the players' intrinsic characteristics.

Concerning the relationship between balance and sport performance, even though COP velocity ($MVM_{COP}$) decreased with age, no correlation with tennis serve performance was found in the whole sample. However, as $DFA_V$ showed a moderate correlation to serve performance (accuracy and ball velocity). From the authors' point of view, these results reinforce the

aforementioned statement that balance performance parameters are not sensitive enough to discriminate between participants according to their performance level while the analysis of COP complexity would be a better approach.

When analyzing correlations in each subgroup, the effect of the *expertise* and *age* in the whole sample, previously remarked in the ANOVA analysis, is faded. A marginal positive correlation between error (MRE) and COP velocity ($MVM_{COP}$) was only found in expert U16 players, but the small subgroup sample could bias it. Moreover, being $DFA_V$ the main variable discriminating balance performance between expert and recreational players, the correlational analysis in each subgroup did not show relation between $DFA_V$ and performance. From the authors' point of view, although the ability of performing higher number of postural adjustments seems to be a differential feature of expert tennis players, this ability would not be crucial in the tennis serve performance. Nevertheless, this does not reject other potential balance influences on other drills related to tennis performance as fast cutting maneuvers, sudden decelerations, and pivoting maneuvers.

This study presents the inherent limitations of a cross-sectional and correlational design, and thus longitudinal and experimental studies should be carried out to obtain a better understanding of balance, age and expertise level relationship. In addition, although we measured 106 participants, the subgroup sample sizes could have biased the correlational analyses as the parameters, like anthropometric or physical features, may have confounded the correlational results. Finally, another limitation to consider is the lack of control about the training program the players were following, apart from the specific tennis training. Any other fitness training performed by the participants could also cause an improvement in balance. Thus, future studies should include this information.

## Conclusions

$DFA_V$ of COP was more sensitive to discriminate between expert and recreational tennis players than balance body sway variables, supporting the idea that, along the years, individuals with higher level of expertise exhibit balance adaptations which are characterized by a higher ability to perform motor adjustments. Although the postural adjustments during a balance task seem to be a differential feature of expert tennis players, this ability does not seem to be crucial in the tennis serve performance. Nevertheless, this does not reject other potential relationships between balance and other drills related to tennis performance as fast cutting maneuvers, sudden decelerations, and pivoting maneuvers.

## Supporting information

**S1 Database. MRE = Mean Radial Error; $MVM_{COP}$ = Mean of the Velocity Magnitude of the Center of Pressure; DFAv = Detrended Fluctuation Analysis from the velocity magnitude time series.**
(XLSX)

## Acknowledgments

This study was made possible by the collaboration of the Area of Teaching and Research of the Royal Spanish Tennis Federation (RFET).

## Author Contributions

**Conceptualization:** Carla Caballero, David Barbado, Francisco J. Moreno.

**Data curation:** Carla Caballero, Héctor Hérnandez-Davó, José Luis Hernández-Davó.

**Formal analysis:** Carla Caballero, Héctor Hérnandez-Davó, José Luis Hernández-Davó.

**Funding acquisition:** Francisco J. Moreno.

**Investigation:** Carla Caballero, David Barbado, Héctor Hérnandez-Davó, José Luis Hernández-Davó.

**Methodology:** Carla Caballero, Francisco J. Moreno.

**Project administration:** Francisco J. Moreno.

**Resources:** Francisco J. Moreno.

**Software:** David Barbado.

**Supervision:** David Barbado, Francisco J. Moreno.

**Validation:** Francisco J. Moreno.

**Visualization:** Francisco J. Moreno.

**Writing – original draft:** Carla Caballero.

**Writing – review & editing:** Carla Caballero, David Barbado, Héctor Hérnandez-Davó, José Luis Hernández-Davó, Francisco J. Moreno.

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
