## [Decision Letter · Decision Letter 0]

16 Nov 2020

PONE-D-20-23789

Balance dynamics are related to maturation and motor dexterity. Application in young and adult tennis players

PLOS ONE

Dear Dr. Moreno,

Thank you for submitting your manuscript to PLOS ONE. After careful consideration, we feel that it has merit but does not fully meet PLOS ONE’s publication criteria as it currently stands. Therefore, we invite you to submit a revised version of the manuscript that addresses the points raised during the review process.

The manuscript had two indications for major revisions and one for rejection. I consider that you can address the comments indicated by reviewers. Please pay attention to the method and results section. It is necessary for more details and a clear presentation. Also, please consider all comments even of the reviewer that rejected the manuscript. 

We look forward to receiving your revised manuscript.

Kind regards,

Fabio A. Barbieri, PhD

Academic Editor

PLOS ONE

Journal Requirements:

'This study was made possible by financial support from the Economy, Industry and Competitiveness Ministry of Spain, projects cod DEP2013-44160-P and DEP2016- 79395-P, Spanish Government. We also thank the collaboration of the Area of Teaching and Research of the Royal Spanish Tennis Federation (RFET).'

'The funders had no role in study design, data collection and analysis, decision to publish, or preparation of the manuscript.'

b. Please include in your financial disclosure statement the name of the funders of this study (as well as grant numbers if available). At present, this information is only available in your acknowledgement section.

4. Please provide further details regarding how participants were recruited, including the participant recruitment date.

5. Please state the inclusion and exclusion criteria used during participant recruitment.

6. You indicated that you had ethical approval for your study.

In your Methods section, please ensure you have also stated whether you obtained consent from parents or guardians of the minors included in the study or whether the research ethics committee or IRB specifically waived the need for their consent.

Reviewers' comments:

Reviewer's Responses to Questions

**Comments to the Author**

1. Is the manuscript technically sound, and do the data support the conclusions?

Reviewer #1: Partly

Reviewer #2: Partly

Reviewer #3: Partly

2. Has the statistical analysis been performed appropriately and rigorously? 

Reviewer #1: Yes

Reviewer #2: Yes

Reviewer #3: No

3. Have the authors made all data underlying the findings in their manuscript fully available?

Reviewer #1: Yes

Reviewer #2: Yes

Reviewer #3: Yes

4. Is the manuscript presented in an intelligible fashion and written in standard English?

Reviewer #1: Yes

Reviewer #2: Yes

Reviewer #3: No

5. Review Comments to the Author

Reviewer #1: An ability is a genetically trait, thought not to be modifiable by practice. I believe the authors use the word ability when they mean the word skill. Bachman, John (RQ, 1961) shows that there seems not to be a balance ability. The authors should scour the old literature on balance.

The observation that DFAv is able to differentiate skilled and less skilled tennis players, by itself does not seem important enough.

It is not clear what the underlying mechanisms, or dynamical systems processes are being influenced by DFA, while the historical standard measurements are insensitive.

Furthermore serving a tennis ball does not seem to be much of a balance task.

Reviewer #2: This paper examined the relationships between balance and motor performance of tennis players of different ages and expertise. They found that balance adaptation, quantified using a detrended fluctuation analysis of the center of pressure, was sensitive enough to detect differences in age for expert players, but the mean of the center of pressure velocity could not. There was no relationship between any balance measures and tennis serve performance. They conclude that sports experience can lead to balance adaptation and that more traditional linear scattering variables of balance are not sensitive enough to reveal subtle changes in balance.

Overall, the paper reveals interesting relationships between balance adaptation and tennis experience (combination of age and expertise). The methods lacks some details and the presentation of the results could be more effective and transparent with plots instead of tables. Also, the authors should not infer any directional effects of tennis experience on balance and vice versa in their conclusions.

The authors need to avoid directional conclusions about whether tennis experience improves balance adaptation or whether better balance improves tennis. The authors have revealed a relationship between balance adaptation and tennis experience, not a directional effect. Lines 77-78 in the introduction refer to one direction while the conclusions in lines 348-350 are framed in the opposite direction. I think the authors know that they cannot assess directionality as the aims stated in lines 103-108 are phrased in a non-directional manner.

-Lines 73-74, "In a bidirectional way, it would also be expected that tennis experience induces balance adaptations. Nevertheless, there is little evidence confirming both assumptions."

-Lines 77-78, "no study has assessed the potential influence of balance on tennis performance."

-Lines 297-298, "tennis experience seems to induce balance adaptations characterized by a higher ability to perform motor adjustments."

-Lines 348-350: "Balance adaptation induced by tennis practice seems to be related with a higher ability to perform postural adjustments."

A factor that the authors do not discuss is that players 18+ who are classified as expert are likely also running, cross-training, lifting weights, doing plyometrics, etc, which would also improve balance. Without controlling or accounting for fitness programs that the older expert players are likely doing, the authors should refrain from concluding that it is tennis alone that is leading to the balance adaptation for expert players. This should be added to the limitations and/or discussion.

The authors are strongly encouraged to present the results in Table 2 as a figure

The authors need to provide scatter plots of the correlated variables so show that the relationship is linear, which is an assumption of Pearson Product-Moment Correlations.

The challenging unstable balance task is not described (lines 139-142). What is an “unstable surface?” The participants are standing on a force platform, which is not unstable on its own.

How is the COP mean of the velocity magnitude computed? COP is a position variable, not velocity, so the authors must have computed COP velocity.

Is there an upper limit on age for the +18 group? Would all subjects be “young,” < 30 years old?

What statistical software was used to perform the statistics?

The authors are encouraged to consider either writing out MVM as mean of the velocity magnitude and MRE as mean radial error, or perhaps using a simpler term that can be written out compared to using a specific abbreviation? I had to frequently remind myself what these abbreviations meant. DFAv is a more acceptable as it is more commonplace.

Line 187 – what is N?

Line 206, should it be “(expert and recreational) as the between-subjects’ factors.”

Line 233 – does older groups mean U16 and +18? Writing it out would be clearer.

Values < 1 are easier to see with 0’s in front of the decimal, ex. 0.12 versus .12. Consider adding zeros before decimals in the tables and text.

Table 2, is there a difference between “Expertise effect” in the MRE row and “Expertise level” used in the other rows?

Tables 3 & 4, please provide an explanation of bolded values

Tables 4 & 5, the DFAv rows seem unnecessary as no values are reported in that row. Please remove.

In Table 5, there are no ** so “** p < .01” can be removed.

Reviewer #3: I. General Comments:

The purpose of this study was to assess the relationship between balance and tennis performance. Specifically, the purposes were: 1) to compare the balance of tennis players of different ages and levels of expertise, and 2) to analyze the relationship between balance and tennis performance (serving speed and accuracy). This subject is relevant to the fields of motor behavior. However, there are some issues which warrant clarification. Specific comments are displayed below.

II. Specific Comments:

(1) In the Abstract section the authors said that “This study compares different cohorts of tennis players with different expertise levels and different ages to analyze how balance performance and its dynamics can determine sport expertise” (lines 25-28). Was the purpose of the study really that? Based upon the data obtained, is it possible to analyze how balance dynamics determines sport expertise?

(2) Different fonts were used in some parts of the manuscript (e.g., lines 48 and 72).

(3) The concept of “balance” must be clarified. What do “balance tasks” mean (line 99)? What exactly do the authors mean? Doesn’t every movement performed involve balance?

(4) The term “maturation” is used in the title and throughout the manuscript (e.g., lines 101, 275, 336). How was maturation evaluated?! According to Methods section, participants were categorized by age (lines 119-120). Therefore, it seems that the authors refer to age as maturation, thereby inferring age to be the same as maturation. Is this correct?

(5) How many trials were performed in the balance protocol? In the Data Analysis section, it is said that “the first 10s of each trial were discarded…” (line 169) but there is no mention about the number of trials.

(6) Results revealed no Interaction (Age x Expertise) for service accuracy. So, why did the authors claim that “expert players significantly improved their serve accuracy (MRE) with age, while recreational players did not show significant differences according to age ranges” (lines 225-228). If there was no interaction, the multiple pair comparisons do not matter.

(7) When the purpose of the study is mentioned in the Discussion section, the author replaced “levels of expertise” by “levels of dexterity” (lines 258-261). Are these terms (expertise and dexterity) synonyms?

(8) Discussion section is not so clear. There is no proper connection among some paragraphs and others are unnecessary. The authors should rewrite this section and provide a coherent, logical framework that actually discuss the relevant literature in terms of its findings and then present a justified conclusion.

6. PLOS authors have the option to publish the peer review history of their article (what does this mean?). If published, this will include your full peer review and any attached files.

Reviewer #1: **Yes: **Howard Zelaznik

Reviewer #2: No

Reviewer #3: No

---

## [Author Response · Author response to Decision Letter 0]

15 Dec 2020

The authors would like to thank you for your advice and recommendations. 

We have performed a revision of this manuscript based on your suggestions. 

In addition, the manuscript has been submitted to Altair K. Fanto Proof-Reading-Service (NIF: X0692378D; C/ La Goleta, 6; C.P.: 03540, Alicante, Spain) for editing and proofreading.

All the changes made are in red in the "manuscript with track changes" file and each reviewer’s comments has been responded on a point-by-point basis in the file "Response to Reviewers":

The response to reviewers is retyped below.

Journal Requirements:

and

Authors’ response: All the style requirements have been met.

'This study was made possible by financial support from the Economy, Industry and Competitiveness Ministry of Spain, projects cod DEP2013-44160-P and DEP2016- 79395-P, Spanish Government. We also thank the collaboration of the Area of Teaching and Research of the Royal Spanish Tennis Federation (RFET).'

'The funders had no role in study design, data collection and analysis, decision to publish, or preparation of the manuscript.'

Authors’ response: The Funding Information has been removed from the Acknowledgements section. We have updated the Funding Statement. Thank you.

b. Please include in your financial disclosure statement the name of the funders of this study (as well as grant numbers if available). At present, this information is only available in your acknowledgement section.

Authors’ response: All the financial information has been updated.

 Authors’ response: The statement has been included.

3. Please include captions for your Supporting Information files at the end of your manuscript, and update any in-text citations to match accordingly. Please see our Supporting Information guidelines for more information: 

http://journals.plos.org/plosone/s/supporting-information

Authors’ response: The caption for the Supporting Information has been included and cited in the Results section. 

Lines 224-225: “The whole database can be found in the Supporting Information (S1. Database)”.

4. Please provide further details regarding how participants were recruited, including the participant recruitment date.

Authors’ response: The information required by the reviewer has been added in the Participants section. Please, see bellow: 

Lines 117-123: “Fifty-five of them were expert players recruited from the Royal Spanish Tennis Federation during the 2012-2013 sport season. In order to be recruited, they had to meet the following criteria: to train at least four times per week; and to play in national or international competitions. In addition, fifty-one participants were recruited as recreational players in the same season from the university where the study was conducted. The recreational players met the following criteria: i) to train no more than two times per week; and to have not taken part in any national or international competition”.

5. Please state the inclusion and exclusion criteria used during participant recruitment.

Authors’ response: The information required by the reviewer has been added in the Participants section. Please, see the answer to the previous comment.

6. You indicated that you had ethical approval for your study.

In your Methods section, please ensure you have also stated whether you obtained consent from parents or guardians of the minors included in the study or whether the research ethics committee or IRB specifically waived the need for their consent.

Authors’ response: The information about the consent from parents or guardians of the minors has been included in the Participants section. 

Lines 127-129: “For both U12 (under 12 years old) and U16 (under 16 years old) groups, written informed consent was signed by the players’ guardians”.

 

Reviewers' comments:

Reviewer's Responses to Questions

Comments to the Author

Reviewer #1: 

 An ability is a genetically trait, though not to be modifiable by practice. I believe the authors use the word ability when they mean the word skill. Bachman, John (RQ, 1961) shows that there seems not to be a balance ability. The authors should scour the old literature on balance.

Authors’ response: The authors understand the reviewer's concern about the controversial meaning of "ability." We followed the definition used by several authors such as Hrysomallis (the body's ability to maintain its center of gravity above its base of support"), or Gerbino Griffin and Zurakowski ("the ability to stand with as little sway as possible"). From these definitions, an ability is not a genetic trait, but each individual's genetic features certainly influence it.

 Hrysomallis, C. (2011). Balance ability and athletic performance. Sports Medicine, 41(3), 221-232. 

 Gerbino, P. G., Griffin, E. D., & Zurakowski, D. (2007). Comparison of standing balance between female collegiate dancers and soccer players. Gait & Posture, 26(4), 501-507.

In addition, when we used the term “balance ability” in a scientific searching database such as PubMed, 20.306 results were found. However, when the “balance skill” was used, 5.850 results were found. (Searching date: 12/02/2020).

Nevertheless, we understand that several authors considered the term ability as “a general trait or capacity of an individual that underlies the performance of a variety of movement skills” (Burton & Miller, 1998). Thus, the term “balance ability” has been changed by “balance”.

 Burton, A.W., & Miller, D.E. (1998). Movement skill assessment. Champaign, IL: Human Kinetics.

 The observation that DFAv is able to differentiate skilled and less skilled tennis players, by itself does not seem important enough. 

 It is not clear what the underlying mechanisms, or dynamical systems processes are being influenced by DFA, while the historical standard measurements are insensitive.

Authors’ response: The authors understand the reviewer’s concern about the usefulness of detrended fluctuation analysis (DFA) as an index of potential control underlying mechanisms rather than only discriminating between groups. The DFA is a mathematical tool that quantifies the long-range auto-correlation of a signal. This is, DFA assesses the extent to which further part of a signal can be predicted by the previous information contained in that signal. Therefore, from the motor behavior perspective, this parameter quantifies the extent to which further motor behavior depends on previous motor fluctuations. Based on this interpretation, several works have linked the information obtained from this parameter with the number of postural changes performed during a motor task. Lower DFA scores mean less dependence on previous behavior (less persistent behavior), which indicates a higher probability of modifying the previous tendency of the movement. These DFA scores have been interpreted as higher flexibility to perform motion adjustments (Amoud et al., 2007; Wang & Yang, 2012). Some studies on balance tasks in older (Manor et al., 2010; Zhou et al., 2013) and young individuals (Barbado et al., 2012) seem to confirm this interpretation as they revealed that individuals who showed lower long-range auto-correlation of the center of pressure (COP) fluctuations while standing on a stable surface demonstrated better performance with more difficult balance tasks. Lower DFA scores (“higher ability to perform motion adjustments”) have also been related to higher learning rates (Barbado et al., 2017). There are some potential underlying mechanisms related to a higher or lower ability to perform motion adjustments. DFA scores and their interpretation as “higher or lower amount of postural or motion adjustments” have been related to some underlying mechanism as the individuals’ sensitivity to their own motor error. DFA scores also depend on other individual factors, such as effort, motivation, or attention. For example, Correll (2008) observed that higher effort was associated with a lower auto-correlated time response variability (low DFA scores) during a “decision-making shooting task,” which was interpreted as a high implication or effort to perform motion adjustment to reduce motor output error. Therefore, analyzing tennis players’ balance behavior through non-linear parameters would help to reveal if tennis experience is related to higher flexibility to perform postural adjustment during balance tasks.

 Amoud, H., Abadi, M., Hewson, D. J., Michel-Pellegrino, V., Doussot, M., & Duchene, J. (2007). Fractal time series analysis of postural stability in elderly and control subjects. Journal of Neuroengineering and Rehabilitation, 4, 12-24.

 Barbado, D., Sabido, R., Vera-Garcia, F. J., Gusi, N., & Moreno, F. J. (2012). Effect of increasing difficulty in standing balance tasks with visual feedback on postural sway and EMG: complexity and performance. Human Movement Science, 31(5), 1224-1237.

 Barbado D, Caballero C, Moreside JM, Vera-García FJ, Moreno FJ. Can be the structure of motor variability predict learning rate? Journal of Experimental Psychology: Human Perception and Performance. 2017;43:596-607.

 Correll, J. (2008). 1/f noise and effort on implicit measures of bias. Journal of Personality and Social Psychology, 94(1), 48-59.

 Manor, B., Costa, M. D., Hu, K., Newton, E., Starobinets, O., Kang, H. G., . . . Lipsitz, L. A. (2010). Physiological complexity and system adaptability: evidence from postural control dynamics of older adults. Journal of Applied Physiology, 109(6), 1786-1791.

 Wang, C. C., & Yang, W. H. (2012). Using detrended fluctuation analysis (DFA) to analyze whether vibratory insoles enhance balance stability for elderly fallers. Archives of Gerontology and Geriatrics, 55(3), 673-676.

 Zhou, J., Manor, B., Liu, D., Hu, K., Zhang, J., & Fang, J. (2013). The complexity of standing postural control in older adults: a modified detrended fluctuation analysis based upon the empirical mode decomposition algorithm. PLoS ONE, 8(5), 1-7.

Based on the above-mentioned rationale, we have performed some modifications along the manuscript to reinforce the physiological interpretation of DFA scores observed in our study. Please, see below:

Lines 91-98: “Specifically, non-linear parameters (e.g., detrended fluctuation analysis), which analyze how motor behavior changes over time, have been linked to some potential underlying mechanisms related to a higher or lower ability to perform motion adjustments (23, 24). For example, DFA scores have been related to the individuals’ sensitivity to their own motor error (21) or the effort to accomplish a motor task (23). Analyzing tennis players’ balance behavior through non-linear parameters would help to reveal if tennis experience is related to higher flexibility to perform postural adjustment during balance tasks”.

Lines 290-302: “Therefore, these results reinforce that DFAV was more sensitive than balance body sway parameters to quantify how motor behavior changes over time, providing information about changes in postural control and the system ability to adapt that linear variables cannot identify (21, 22). According to several works, the low DFAV in the COP excursion revealed that expert and oldest tennis players performed a higher number of movement adjustments during the balance task (23). These results could indicate that these participants displayed a more exploratory behavior (44) or higher sensitivity to their own motor error (21), allowing them to obtain better performance and facilitating adaptation processes (21, 43). The fact that only expert players showed a reduction in DFAV scores with age while no changes were observed in the recreational tennis players would reinforce that tennis experience rather than age seems to be related to a better postural control system, characterized by a higher ability to perform motor adjustments”.

 Furthermore, serving a tennis ball does not seem to be much of a balance task.

Authors’ response: The authors understand the reviewer’s concern about the “apparently” lack of relationship between tennis serve and balance tasks. Probably, other tasks, as the change of direction drills, could be more related to balance performance. However, serving a tennis ball is one of the most important skills for tennis success, and, at the same time, balance has been considered by coaches as one of the most important determinants of serve success (even though there is no clear link between tennis serve and balance). The scientific literature clearly supports the idea that balance is related to athletic performance in those sports or tasks in which balance is a clear requirement, such as in ski or gymnastic skills. However, the question is if balance can be a determinant factor for general sports skills. That is what the authors wanted to answer.

In order to clarify why tennis serve was used in this study, the authors have added some explanations at the end of the Introduction section. Please, see below:

Lines 110-112: “Tennis performance was assessed through the accuracy of the tennis serve as a way to analyze a tennis skill very relevant to tennis success but not directly related to balance control”.

Reviewer #2: 

This paper examined the relationships between balance and motor performance of tennis players of different ages and expertise. They found that balance adaptation, quantified using a detrended fluctuation analysis of the center of pressure, was sensitive enough to detect differences in age for expert players, but the mean of the center of pressure velocity could not. There was no relationship between any balance measures and tennis serve performance. They conclude that sports experience can lead to balance adaptation and that more traditional linear scattering variables of balance are not sensitive enough to reveal subtle changes in balance.

Overall, the paper reveals interesting relationships between balance adaptation and tennis experience (combination of age and expertise). The methods lack some details, and the presentation of the results could be more effective and transparent with plots instead of tables. Also, the authors should not infer any directional effects of tennis experience on balance and vice versa in their conclusions.

 The authors need to avoid directional conclusions about whether tennis experience improves balance adaptation or whether better balance improves tennis. The authors have revealed a relationship between balance adaptation and tennis experience, not a directional effect. Lines 77-78 in the introduction refer to one direction while the conclusions in lines 348-350 are framed in the opposite direction. I think the authors know that they cannot assess directionality as the aims stated in lines 103-108 are phrased in a non-directional manner.

-Lines 73-74, "In a bidirectional way, it would also be expected that tennis experience induces balance adaptations. Nevertheless, there is little evidence confirming both assumptions."

-Lines 77-78, "no study has assessed the potential influence of balance on tennis performance."

-Lines 297-298, "tennis experience seems to induce balance adaptations characterized by a higher ability to perform motor adjustments."

-Lines 348-350: "Balance adaptation induced by tennis practice seems to be related with a higher ability to perform postural adjustments."

Authors’ response: The authors agree with the reviewer. This work is a correlational study, so a cause-effect relationship cannot be found. That is a limitation of the study that has been added at the end of the Discussion (line 329). In addition, the writing has been checked in order to avoid directional conclusions:

Line 68: "In a bidirectional way, it would also be expected that tennis experience induces balance adaptations. Nevertheless, there is little evidence confirming both assumptions" has been changed by “In addition, it would also be expected that tennis experience induces balance adaptations. Nevertheless, there is little evidence confirming both assumptions”.

Lines 72-73: “no study has assessed the potential influence of balance on tennis performance” has been changed by “no study has assessed the potential relationship between balance and tennis performance”

Lines 300-302: "tennis experience seems to induce balance adaptations characterized by a higher ability to perform motor adjustments" has been changed by “tennis experience rather than age seems to be related to a better postural control system, characterized by a higher ability to perform motor adjustments”.

“Balance adaptation induced by tennis practice seems to be related with a higher ability to perform postural adjustments” has been removed due to the changes the authors have made in this section.

 A factor that the authors do not discuss is that players 18+ who are classified as expert are likely also running, cross-training, lifting weights, doing plyometrics, etc, which would also improve balance. Without controlling or accounting for fitness programs that the older expert players are likely doing, the authors should refrain from concluding that it is tennis alone that is leading to the balance adaptation for expert players. This should be added to the limitations and/or discussion.

Authors’ response: The authors understand the reviewer’s concern. The limitation of no controlling the fitness programs participants were doing has been added at the end of the discussion as a limitation of the study. 

Lines 334-337: “Finally, another limitation to consider is the lack of control about the training program the players were following, apart from the specific tennis training. Any other fitness training performed by the participants could also cause an improvement in balance. Thus, future studies should include this information”.

 The authors are strongly encouraged to present the results in Table 2 as a figure.

Authors’ response: Thank you very much for the suggestion. The Table 2 has been presented as a figure. Please, see below:

Figure 2. Pair comparisons between age ranges and expertise levels. MRE = Medial Radial Error; BVE = Bivariate Variable Error; MVM = Mean Velocity Magnitude; DFAV = Detrended Fluctuation Analysis of COP time series velocity data.

* Significant differences compared to U12; # Significant differences compared to U16; † Significant differences between levels.

 The authors need to provide scatter plots of the correlated variables so show that the relationship is linear, which is an assumption of Pearson Product-Moment Correlations.

Authors’ response: This information has not been added to the manuscript due to its dimensions. Please, check Appendix 1 to see the scatter plots of all the correlations that are presented in the study.

 The challenging unstable balance task is not described (lines 139-142). What is an “unstable surface?” The participants are standing on a force platform, which is not unstable on its own.

Authors’ response: The information about the unstable surface has been added in the Experimental Procedure and Data Collection section. Please, see below:

Lines 139-142: “The unstable surface consisted of a rigid wooden platform (diameter: 55 cm) affixed to the flat surface of a polyester resin hemisphere (diameter of the hemisphere: 35 cm; height of the platform relative to the bottom of the hemisphere: 12 cm).”

Lines 146-148: “In order to assess postural stability, the unstable surface was placed on a force platform. Ground reaction forces were recorded (1000 Hz) to analyze the center of pressure (COP) data that were subsampled at 20 Hz to avoid artificial co-linearities (30)”.

 How is the COP mean of the velocity magnitude computed? COP is a position variable, not velocity, so the authors must have computed COP velocity.

Authors’ response: Mean magnitude velocity was calculated according to Prieto, Myklebust, Hoffmann, Lovett, & Myklebust (1996) as the total length of the COP path (i.e., sum of the distances between consecutive points on the COP path) normalized by the duration of the tasks.

(∑_(n+1)^(N-1)▒√(〖〖〖(AP〗_(n+1)- AP〗_n)〗^2+ 〖〖〖(ML〗_(n+1)- ML〗_n)〗^2 ))/T

Where N is the number of data point of the time series, n is a specific point of the data series, AP is the COP position in the anterior-posterior axis, ML is the COP position in the medial-lateral axis, and T is the total duration (seconds) of a task.

 Prieto TE, Myklebust JB, Hoffmann RG, Lovett EG, Myklebust BM. Measures of postural steadiness: differences between healthy young and elderly adults. IEEE Trans Biomed Eng. 1996;43(9):956-66

The references paper has been cited in reviewed manuscript (Data Analysis and Reduction Section, line 177).

 Is there an upper limit on age for the +18 group? Would all subjects be “young,” < 30 years old?

 Authors’ response: The oldest tennis player who was measured was included in the expert group and he was 35 years old. There was no upper limit on age because the players were selected according to the Royal Spanish Tennis Federation.

 What statistical software was used to perform the statistics?

Authors’ response: The statistical analysis was performed using IBM SPSS Statistics 22.0.0.0. This information has been added to the Statistical Analysis section. 

Lines 218-219: “Statistical tests were made using IBM SPSS Statistics 22.0.0.0. The alpha value of significance effect was set at p < .05.”

 The authors are encouraged to consider either writing out MVM as mean of the velocity magnitude and MRE as mean radial error, or perhaps using a simpler term that can be written out compared to using a specific abbreviation? I had to frequently remind myself what these abbreviations meant. DFAv is a more acceptable as it is more commonplace.

Authors’ response: The authors understand the concern about the ease of reading the manuscript. The terms MVM and MRE have been kept in the result section since they are frequently used in the literature but changed them by simpler terms in the discussion section to help the understanding of the text.

 

 Line 187 – what is N?

Authors’ response: N is the number of data points of the time series. We have clarified it in the manuscript. Please, see below:

Lines 192-193: “In this study, the slope α was obtained from the window range 4 ≤ n ≤ N/10 to maximize the long-range correlations and reduce errors incurred by estimating α [38], being N the total number of the time series data point.”

 Line 206, should it be “(expert and recreational) as the between-subjects’ factors.”

Authors’ response: Line 206 has been changed following the recommendation. 

 Line 233 – does older groups mean U16 and +18? Writing it out would be clearer.

Authors’ response: Older groups meant both the recreational and expert groups +18. We have clarified it in the manuscript. 

Lines 238-240: “Concerning the balance task, main effect of age was found, showing a reduction in both MVMCOP (F2,105 = 14.378; p < 0.001; ƞ_p^2= 0.215) and DFAV (F2,105 = 3.420; p = 0.036; ƞ_p^2= 0.061) in the +18 group, both expert and recreational players (Fig 2)”.

 Values < 1 are easier to see with 0’s in front of the decimal, ex. 0.12 versus .12. Consider adding zeros before decimals in the tables and text.

Authors’ response: All the decimal numbers have been changed adding the 0’s before decimals.

 Table 2, is there a difference between “Expertise effect” in the MRE row and “Expertise level” used in the other rows?

Authors’ response: There is no difference. It was a mistake that has been fixed. Thank you very much for letting the authors know.

 Tables 3 & 4, please provide an explanation of bolded values

Authors’ response: The bold font has been used to improve the visualization for all the significant correlations. This information has been added to the legend of the tables.

 Tables 4 & 5, the DFAv rows seem unnecessary as no values are reported in that row. Please remove.

Authors’ response: The DFAv rows have been removed. It was a mistake. Thank you very much for letting the authors know.

 In Table 5, there are no ** so “** p < .01” can be removed.

Authors’ response: Thank you very much for realizing it. The p < .01 has been removed.

Reviewer #3: 

I. General Comments:

The purpose of this study was to assess the relationship between balance and tennis performance. Specifically, the purposes were: 1) to compare the balance of tennis players of different ages and levels of expertise, and 2) to analyze the relationship between balance and tennis performance (serving speed and accuracy). This subject is relevant to the fields of motor behavior. However, there are some issues which warrant clarification. Specific comments are displayed below.

II. Specific Comments:

(1) In the Abstract section the authors said that “This study compares different cohorts of tennis players with different expertise levels and different ages to analyze how balance performance and its dynamics can determine sport expertise” (lines 25-28). Was the purpose of the study really that? Based upon the data obtained, is it possible to analyze how balance dynamics determines sport expertise?

Authors’ response: The authors agree with the reviewer's comment. The introduction and aim of the study in the abstract have been modified. Please, see below: 

Lines 17-23: 

“In tennis, coaches consider balance as a fundamental ability for the acquisition of skilled motor performance. However, the potential relationship between balance and tennis expertise and performance has not been explored yet. Therefore, this study assessed the relationship between balance ability and tennis performance using linear and non-linear parameters through 1) the comparison of tennis players of different ages and levels of expertise, and 2) analyzing the relationship between balance and tennis serving speed and accuracy.”

(2) Different fonts were used in some parts of the manuscript (e.g., lines 48 and 72).

Authors’ response: The mistake has been fixed.

(3) The concept of “balance” must be clarified. What do “balance tasks” mean (line 99)? What exactly do the authors mean? Doesn’t every movement performed involve balance?

Authors’ response: The term balance has been clarified in the introduction section. 

Lines 41-43: “Balance control requires to maintain the position of the body’s center of gravity vertically over the base of support (1,2)”. 

The authors agree with the reviewer that every movement performed involves balance. However, in the literature, the term balance task usually refers to a task that requires an effort to maintain the body’s center of gravity vertically over the base of support, trying to display minimal movement. This requirement is explained in the Experimental Procedure and Data Collection section. 

Lines 139: “Participants were asked to stand ‘as still as possible’ on an unstable surface”.

 Nashner LM. Practical biomechanics and physiology of balance. In: Jacobson GP, Newman, C.W. & Kartush, J.M., editor. Handbook of balance function testing. San Diego (CA): Singular Publishing Group; 1997. p. 261-79.

 Hrysomallis C. Balance ability and athletic performance. Sports Med. 2011;41(3):221-32.

(4) The term “maturation” is used in the title and throughout the manuscript (e.g., lines 101, 275, 336). How was maturation evaluated?! According to Methods section, participants were categorized by age (lines 119-120). Therefore, it seems that the authors refer to age as maturation, thereby inferring age to be the same as maturation. Is this correct?

Authors’ response: The authors understand the reviewer’s concern and agree with the fact that “maturation” is related to age but it is not the same. It is true that both terms were used along the manuscript as synonymous, but it has been changed using only the term age, since it is the one we have registered.

(5) How many trials were performed in the balance protocol? In the Data Analysis section, it is said that “the first 10s of each trial were discarded…” (line 169) but there is no mention about the number of trials.

Authors’ response: The participants performed the balance protocol one, keeping their balance for 60 s. This information has been added to the Experimental Procedure and Data Collection section.

Lines 145-146: “Participants performed one trial keeping this position for 60 seconds”.

(6) Results revealed no Interaction (Age x Expertise) for service accuracy. So, why did the authors claim that “expert players significantly improved their serve accuracy (MRE) with age, while recreational players did not show significant differences according to age ranges” (lines 225-228). If there was no interaction, the multiple pair comparisons do not matter.

Authors’ response: The authors completely agree that with the results obtained it cannot be said that affirmation. There is a trend that could reflect the affirmation but it is not statistically proved due to the reduction of the sample once it is split according to the age and skill level. The sentence has been rephrased.

Lines 231-233: “multiple pair comparisons showed that older expert players (+18) significantly displayed better performance in their serve accuracy (MRE) than the youngest expert players (U12)”.

(7) When the purpose of the study is mentioned in the Discussion section, the author replaced “levels of expertise” by “levels of dexterity” (lines 258-261). Are these terms (expertise and dexterity) synonyms?

Authors’ response: The authors have considered both terms as synonyms because they have been used referring to the aptitude or skill topic. However, in order to ease the reading of the manuscript, the term ‘levels of expertise’ has been unified along the whole manuscript.

(8) Discussion section is not so clear. There is no proper connection among some paragraphs and others are unnecessary. The authors should rewrite this section and provide a coherent, logical framework that actually discuss the relevant literature in terms of its findings and then present a justified conclusion.

Authors’ response: The Discussion section has been modified in order to accomplish all the previous comments and also to improve the coherence and connection between the paragraphs. Some paragraphs have been removed, and others have been rewritten. The rewritten parts can be check below:

Lines 281-283: “These results are in line with previous works which did not find any significant effect of sport experience in balance adaptation when traditional postural sway parameters were used, even in sports that are so balance demanding like surfing or judo (10, 18).”

Lines 290-302: “Therefore, these results reinforce that DFAV was more sensitive than balance body sway parameters to quantify how motor behavior changes over time, providing information about changes in postural control and the system ability to adapt that linear variables cannot identify (21, 22). According to several works, the low DFAV in the COP excursion revealed that expert and oldest tennis players performed a higher number of movement adjustments during the balance task (23). These results could indicate that these participants displayed a more exploratory behavior (Wu et al.,) or higher sensitivity to their own motor error (Barbado et al.), allowing them to obtain better performance and facilitating adaptation processes (21, 43). The fact that only expert players showed a reduction in DFAV scores with age while no changes were observed in the recreational tennis players would reinforce that tennis experience rather than age seems to be related to a better postural control system, characterized by a higher ability to perform motor adjustments.”

Lines 303-310: “Regarding the correlational analyses, when the whole sample was assessed, a direct relationship between accuracy and velocity was observed. Previous experiments in tennis serve have found this apparent exception to Fitts’ law (44), so the better tennis players are able to serve faster and more accurately (lower MRE). However, these results are mainly due to the effect of age and expertise, as the ANOVAs showed. The fact that older recreational players did not exhibit significantly greater accuracy than younger players would support that expertise is based not only on age but also on deliberated practice combined with the players' intrinsic characteristics.”

Lines 318-321: “When analyzing correlations in each subgroup, the effect of the expertise and age in the whole sample, previously remarked in the ANOVA analysis, is faded. A marginal positive correlation between error (MRE) and COP velocity (MVMCOP) was only found in expert U16 players, but the small subgroup sample could bias it.”

Lines 334-337: “Finally, another limitation to consider is the lack of control about the training program the players were following, apart from the specific tennis training. Any other fitness training performed by the participants could also cause an improvement in balance. Thus, future studies should include this information.”

Lines 340-343: “DFAV of COP was more sensitive to discriminate between expert and recreational tennis players than balance body sway variables, supporting the idea that, along the years, individuals with higher motor dexterity exhibit balance adaptations which are characterized by a higher ability to perform motor adjustments.”

Lines 346-347: “Nevertheless, this does not reject other potential relationships between balance and other drills related to tennis performance as fast cutting maneuvers, sudden decelerations, and pivoting maneuvers.”

---

## [Decision Letter · Decision Letter 1]

5 Feb 2021

PONE-D-20-23789R1

Balance dynamics are related to age and levels of expertise. Application in young and adult tennis players

PLOS ONE

Dear Dr. Moreno,

Thank you for submitting your manuscript to PLOS ONE. After careful consideration, we feel that it has merit but does not fully meet PLOS ONE’s publication criteria as it currently stands. Therefore, we invite you to submit a revised version of the manuscript that addresses the points raised during the review process.

Dear authors

One reviewer recommended to accept the paper and another one requested some minor revisions. Please addressed the suggestions indicated by the reviewer.

We look forward to receiving your revised manuscript.

Kind regards,

Fabio A. Barbieri, PhD

Academic Editor

PLOS ONE

Reviewers' comments:

Reviewer's Responses to Questions

**Comments to the Author**

1. If the authors have adequately addressed your comments raised in a previous round of review and you feel that this manuscript is now acceptable for publication, you may indicate that here to bypass the “Comments to the Author” section, enter your conflict of interest statement in the “Confidential to Editor” section, and submit your "Accept" recommendation.

Reviewer #2: All comments have been addressed

Reviewer #3: (No Response)

2. Is the manuscript technically sound, and do the data support the conclusions?

Reviewer #2: (No Response)

Reviewer #3: Partly

3. Has the statistical analysis been performed appropriately and rigorously? 

Reviewer #2: (No Response)

Reviewer #3: No

4. Have the authors made all data underlying the findings in their manuscript fully available?

Reviewer #2: (No Response)

Reviewer #3: Yes

5. Is the manuscript presented in an intelligible fashion and written in standard English?

Reviewer #2: (No Response)

Reviewer #3: Yes

6. Review Comments to the Author

Reviewer #2: (No Response)

Reviewer #3: I. General Comments:

The purpose of this study was to assess the relationship between balance and tennis performance. Specifically, the purposes were: 1) to compare the balance of tennis players of different ages and levels of expertise, and 2) to analyze the relationship between balance and tennis performance (serving speed and accuracy). This subject is relevant to the fields of motor behavior. The modifications improved the quality of the manuscript. However, there are still some issues which warrant clarification. Specific comments are displayed below.

II. Specific Comments:

(1) Regarding the concept of “balance”, the authors said that “The term balance has been clarified in the introduction section” (Response to Reviewers file). However, there are no differences between the revised Introduction and the previous one in respect to this issue. The authors just replaced “Balance ability” (line 48 – previous version) by “Balance” (line 41 – present version).

(2) In this reviewed manuscript the authors mentioned the number of trials performed in the balance protocol. According to authors “Participants performed one trial keeping this position for 60 seconds” (lines 145-146). Is one trial enough to provide reliable measurements of CoP? It’s necessary to justify and include the proper references.

(3) Results revealed no Interaction (Age x Expertise) for service accuracy. So, the sentence “…multiple pair comparisons showed that older expert players (+18) significantly displayed better performance in their serve accuracy (MRE) than the youngest expert players (U12), while recreational players did not show significant differences according to age ranges” (lines 225-228) is still statistically unnecessary. If there was no interaction, the multiple pair comparisons do not matter.

7. PLOS authors have the option to publish the peer review history of their article (what does this mean?). If published, this will include your full peer review and any attached files.

Reviewer #2: No

Reviewer #3: No

---

## [Author Response · Author response to Decision Letter 1]

3 Mar 2021

The authors would like to thank you for your advice and recommendations. As you will see, we have performed a revision of this manuscript based on your suggestions.

Reviewers' comments:

Reviewer #3: 

I. General Comments:

The purpose of this study was to assess the relationship between balance and tennis performance. Specifically, the purposes were: 1) to compare the balance of tennis players of different ages and levels of expertise, and 2) to analyze the relationship between balance and tennis performance (serving speed and accuracy). This subject is relevant to the fields of motor behavior. The modifications improved the quality of the manuscript. However, there are still some issues which warrant clarification. Specific comments are displayed below. 

Thank you very much for your recommendations. All the changes made are in red in the manuscript and each reviewer’s comments has been responded on a point-by-point basis:

II. Specific Comments:

1. Regarding the concept of “balance”, the authors said that “The term balance has been clarified in the introduction section” (Response to Reviewers file). However, there are no differences between the revised Introduction and the previous one in respect to this issue. The authors just replaced “Balance ability” (line 48 – previous version) by “Balance” (line 41 – present version).

Thank you for your comment. In the revised version, a definition was added at the beginning of the introduction to clarify the concept of balance control used in the manuscript. The authors have now rephrased the beginning of the introduction section to address more clearly this suggestion. 

Lines 41-47: “Balance is one of the main features of motor control as every human movement involves it. Balance has been defined as the ability of the neuromuscular system to integrate visual, vestibular and somatosensory feedback to control the body’s center of gravity projection (1, 2). Therefore, balance control has been frequently analyzed by the use of tasks that challenge the individual in maintaining the position of the body’s center of gravity vertically over the base of support (1, 2), and it has been commonly assessed through the body’s postural sway (3)”. 

In addition, in the Balance task protocol section, all the details about how the balance task was conducted are explained (Lines 139 – 148).

2. In this reviewed manuscript the authors mentioned the number of trials performed in the balance protocol. According to authors “Participants performed one trial keeping this position for 60 seconds” (lines 145-146). Is one trial enough to provide reliable measurements of CoP? It’s necessary to justify and include the proper references.

Thank you very much for your comment. Some studies have already proved that just 1 trial, even shorter than 60 s, is reliable to measure the COP (Caballero, Barbado & Moreno, 2015). The authors have chosen 60 seconds of measuring because several studies have used this time length in the literature (Caballero et al., 2016; Chapman et al., 2008; Lin, Seol, Nussbaum & Madigan, 2008). The references of the papers have been included in the method section. (Lines 146-148).

Caballero, C., Barbado, D., & Moreno, F. J. (2015). What COP and Kinematic Parameters Better Characterize Postural Control in Standing Balance Tasks? Journal of Motor Behavior, 47(6), 550-562.

Chapman DW, Needham KJ, Allison GT, Lay B, Edwards DJ. Effects of experience in a dynamic environment on postural control. Br J Sports Med. 2008;42(1):16-21.

Lin, D., Seol, H., Nussbaum, M. A., & Madigan, M. L. (2008). Reliability of COP-based postural sway measures and age-related differences. Gait & Posture, 28(2), 337-342.

Santos, B. R., Delisle, A., Lariviere, C., Plamondon, A., & Imbeau, D. (2008). Reliability of centre of pressure summary measures of postural steadiness in healthy young adults. Gait & Posture, 27(3), 408-415.

3. Results revealed no Interaction (Age x Expertise) for service accuracy. So, the sentence “…multiple pair comparisons showed that older expert players (+18) significantly displayed better performance in their serve accuracy (MRE) than the youngest expert players (U12), while recreational players did not show significant differences according to age ranges” (lines 225-228) is still statistically unnecessary. If there was no interaction, the multiple pair comparisons do not matter.

Thank you very much for your comment. The sentence have been removed from the results section.

---

## [Decision Letter · Decision Letter 2]

29 Mar 2021

Balance dynamics are related to age and levels of expertise. Application in young and adult tennis players

PONE-D-20-23789R2

Dear Dr. Moreno,

We’re pleased to inform you that your manuscript has been judged scientifically suitable for publication and will be formally accepted for publication once it meets all outstanding technical requirements.

Kind regards,

Fabio A. Barbieri, PhD

Academic Editor

PLOS ONE

Additional Editor Comments (optional):

Reviewers' comments:

Reviewer's Responses to Questions

**Comments to the Author**

1. If the authors have adequately addressed your comments raised in a previous round of review and you feel that this manuscript is now acceptable for publication, you may indicate that here to bypass the “Comments to the Author” section, enter your conflict of interest statement in the “Confidential to Editor” section, and submit your "Accept" recommendation.

Reviewer #3: All comments have been addressed

2. Is the manuscript technically sound, and do the data support the conclusions?

Reviewer #3: Yes

3. Has the statistical analysis been performed appropriately and rigorously? 

Reviewer #3: Yes

4. Have the authors made all data underlying the findings in their manuscript fully available?

Reviewer #3: Yes

5. Is the manuscript presented in an intelligible fashion and written in standard English?

Reviewer #3: Yes

6. Review Comments to the Author

Reviewer #3: All suggestions made were accepted by the authors. These modifications improved the quality of their manuscript.

7. PLOS authors have the option to publish the peer review history of their article (what does this mean?). If published, this will include your full peer review and any attached files.

Reviewer #3: No

---

## [Editor Report · Acceptance letter]

5 Apr 2021

PONE-D-20-23789R2 

Balance dynamics are related to age and levels of expertise. Application in young and adult tennis players 

Dear Dr. Moreno:

I'm pleased to inform you that your manuscript has been deemed suitable for publication in PLOS ONE. Congratulations! Your manuscript is now with our production department. 

Kind regards, 

on behalf of

Dr. Fabio A. Barbieri 

Academic Editor

PLOS ONE